



# Behaviour of KCl sorbent traps and KCl trapping solutions used for atmospheric mercury speciation: stability and specificity

Jan Gačnik[1,2], Igor Živković[2], Sergio Ribeiro Guevara[3], Radojko Jaćimović[2], Jože Kotnik[2], Gianmarco De Feo[4], Matthew A. Dexter[4], Warren T. Corns[4] and Milena Horvat[1,2]

[1]Jožef Stefan International Postgraduate School, Jamova Cesta 39, 1000 Ljubljana, Slovenia
[2]Department of Environmental Sciences, Jožef Stefan Institute, Jamova Cesta 39, 1000 Ljubljana, Slovenia
[3]Laboratorio de Análisis por Activación Neutrónica, Centro Atómico Bariloche, Av. Bustillo km 9.5, 8400 Bariloche, Argentina
[4]P S Analytical Ltd, Arthur House, Main Road, Orpington, Kent, BR5 3HP, UK

*Correspondence to*: Milena Horvat (milena.horvat@ijs.si)

**Abstract.** Atmospheric mercury speciation is of paramount importance for understanding the behavior of mercury once it is emitted into the atmosphere as gaseous elemental (GEM), gaseous oxidized (GOM) and particulate-bound (PBM) mercury. GOM and PBM sampling are the most problematic steps in the analytical procedure. GOM sampling with speciation traps composed of KCl sorbent materials and KCl trapping solutions are commonly used sampling methods, although the work done

at ambient air concentrations is limited. The results of the specificity test showed that the KCl sorbent traps are very specific when using new traps, while their specificity drops dramatically when they are reused. The results of the stability test showed that the highest $Hg^{2+}$ losses (up to 5.5 % of $Hg^{2+}$ loss) occur when low amounts of $Hg^{2+}$ (< 1 ng) are loaded, due to a reduction of $Hg^{2+}$ to $Hg^0$. GOM losses should be taken into account when using KCl sorbent traps for atmospheric Hg speciation, especially at low ambient GOM concentrations. KCl trapping solutions have also been considered as a selective trapping media

for GOM in atmospheric samples. A dimensionless Henry's law constant was experimentally derived and was used to calculate the solubility of elemental Hg in KCl solution. The degree of GEM oxidation was established by purging elemental Hg calibration gas into a KCl solution and determining the GOM trapped using aqueous phase propylation liquid-liquid extraction GC-AFS. A positive GOM bias was observed due to the solubility and oxidation of GEM in KCl trapping solutions strongly suggesting that this approach is unsuitable for atmospheric mercury speciation measurements.

**1 Introduction**

Since the 19th century, human activities have led to a 450 % increase in the concentration of mercury in the atmosphere. The atmospheric chemistry of mercury and the behaviour of mercury species in the air have been thoroughly studied to gain insight into the global and local mercury cycling (United Nations Environment Programme, 2019). In order to achieve comparability of atmospheric mercury speciation worldwide, the analytical methodology needs to be well understood in the terms of

metrology and the conversion processes that may occur during the analysis. Uncertainties and lack of knowledge already appear in the sampling phase of the analytical procedure (Jaffe et al., 2014; Gustin et al., 2013).





First, the mercury species must be collected from the air and accumulated in a medium suitable for further analysis. Because some species of Hg are present in the atmosphere at very low concentrations, a highly selective pre-concentration step is required to accumulate sufficient quantity of the species for analysis. The collection of total gaseous mercury (TGM) is

achieved by drawing air through different types of quartz traps filled with gold materials at a known and fixed flow rate. Types of gold materials include coiled gold wire, gold nanostructures and high specific surface-area substrate coated with gold (United States Environmental Protection Agency (U.S. EPA), 1999). The decision to use a particular gold material depends primarily on the mass of mercury to be collected. For gaseous oxidized mercury (GOM) the main sampling and preconcentration methods are: KCl-coated denuders (Bu et al., 2018); cation-exchange (CEM), nylon, or

poly(tetrafluoroethylene) (PTFE) membranes (Bu et al., 2018; Huang et al., 2013; Gustin et al., 2021); KCl impinging solutions (impingers, adaptations of the Ontario Hydro method (ASTM International, 2016)) and KCl sorbent traps (U.S. Environmental Protection Agency, 2017; Prestbo and Bloom, 1995). Various types of filters (quartz-fibre, cellulose-acetate, glass-fibre and Teflon filters) are most commonly used for particulate-bound mercury (PBM) sampling (Zhang et al., 2019).

It is generally accepted that all forms of gaseous mercury are collected on gold traps and the measurement represents TGM

(Dumarey, 1985; Shafawi et al., 1999) There is however disagreement whether the sampling protocols applied are optimal for GOM (Gustin and Jaffe, 2010). Most problems with sampling of atmospheric mercury are related to GOM and PBM. Accurate quantification of PBM has proven to be one of the most difficult tasks of atmospheric mercury speciation. Problems include: i) meteorological conditions, adsorption, nucleation, gas-particle partitioning and other physical/chemical processes, ii) ultra-low concentrations of PBM, iii) formation of artifacts during sampling, iv) loss of PBM during longer sampling periods (Wang

et al., 2013). Recently, research has focused on GOM sampling and its dependence on ambient conditions. Various studies of KCl-coated denuders have shown that there is a clear dependence of collection efficiency on relative humidity (McClure et al., 2014; Huang and Gustin, 2015) and on the presence of ozone (McClure et al., 2014; Lyman et al., 2010). The collection efficiency has been found to be inversely dependent on ozone concentration and relative humidity, dropping as low as 13 % in some cases (Wang et al., 2013; McClure et al., 2014; Huang and Gustin, 2015). Experiments using water vapor spikes

showed that not only was the efficiency for GOM collection on KCl-coated denuders very low under high humidity conditions, but also GEM concentrations were increased. This observation made it clear that under conditions of high humidity, GOM can be converted to GEM (or detected as such) (Huang and Gustin, 2015). Underestimated GOM and overestimated GEM due to processes occurring during denuder sampling have been confirmed by Lyman et al. (2016). In addition to denuders, the effectiveness of CEM and nylon membranes at different relative humidity levels was also studied. Humidity had a positive

artifact on GOM for CEM while lowering the collection efficiency of nylon membranes. Nylon membrane passivation occurred with ozone exposure (Huang and Gustin, 2015). Gustin et al. compared different measurement methods for atmospheric Hg speciation. The results showed that GOM and reactive mercury (RM – sum of GOM and PBM), measured with nylon membranes, were approximately half lower than GOM and RM measured with CEM (Gustin et al., 2019). In addition to CEM and nylon membranes, various materials have been tested as membrane materials. CEM and polyethersulfone membrane (PES)

have been shown to be the most quantitative sorbents but they are limited when used for high concentrations of RM/GOM.



Nylon was best for identifying GOM compounds by thermal desorption. Other materials such as anion-exchange membranes, polycarbonate and polypropylene materials have shown unsatisfactory results (Dunham-Cheatham et al., 2020). A three-membrane system (CEM + nylon membranes + PTFE membranes) was recently applied with an attempt to distinguish between PBM and GOM. Authors suggest that PTFE membranes retain mostly PBM while CEM and nylon membranes retain RM
(Gustin et al., 2021).

Various authors have studied the behavior of membranes and denuders as GOM/RM sampling methods, but less work has been done on the behavior of KCl sorbent traps and KCl trapping solutions for mercury atmospheric measurements. Initial experiments on KCl sorbent trap valuation were performed by the authors of the mercury speciation adsorption method (MESA) used for flue gas sampling (Prestbo and Bloom, 1995). The authors determined species stability on KCl sorbent traps
during storage, matrix effects, breakthrough and artifact formation (Prestbo and Bloom, 1995). Although several KCl sorbent trap behavior studies have been performed for flue gas sampling (Electric Power Research Institute (EPRI), 2015), no study has been done on their use for atmospheric mercury speciation measurements. KCl trapping solutions are commonly used for Hg speciation in flue gas (ASTM International, 2016) but the selectivity and suitability for GOM in atmospheric mercury measurements have not been previously investigated. Therefore, the aim of this work was to focus on gathering information
on the behavior of KCl sorbent traps and KCl trapping solutions under different sampling conditions that would complement the aforementioned work and improve knowledge of the processes that may occur during atmospheric Hg speciation. To allow experiments using low Hg amounts (under 1 ng), radioactive [197]Hg tracer can be used as it has been shown to be advantageous in cases where contamination and detection limit are problematic (Ribeiro Guevara et al., 2004; Koron et al., 2012; Ribeiro Guevara and Horvat, 2013).

## 2 Materials and methods

### 2.1 Chemicals and instruments

Chemicals used in this work: 65 % $HNO_3$ (for analysis, Supelco, Darmstadt, Germany), 30 % HCl (Suprapur, Merck, Darmstadt, Germany), 47 % HBr (for analysis, Merck, Darmstadt, Germany), KCl (Suprapur, Merck, Darmstadt, Germany), $SnCl_2 \cdot 2H_2O$ (for analysis, max. 0.000001 % Hg, Merck, Darmstadt, Germany), $HAuCl_4 \cdot xH_2O$ (gold chloride hydrate, 99.995
% trace metal basis, Merck, Darmstadt, Germany) NIST SRM 3133: Mercury (Hg) Standard Solution (National Institute of Standards and Technology, Gaithersburg, MD, USA), [196]Hg enriched elemental Hg (enriched from 0.15 % to 51.58 % [196]Hg, Isoflex, San Francisco, CA, USA), Type I purified water (electrical resistivity 18.2 MΩ cm; Milli-Q water, Merck, Darmstadt, Germany), elemental mercury (99.9999 % Suprapur, Merck, Darmstadt, Germany), sodium tetrapropylborate (Merck, Darmstadt, Germany), acetic acid (puriss., Merck, Darmstadt, Germany), ammonium acetate (LiChropur, Merck, Darmstadt,
Germany), 2,2,4-trimethylpentane (for HPLC, ≥99 %, Merck, Darmstadt, Germany), silica gel (technical grade 40, 6-14 mesh, Merck, Darmstadt, Germany) and anhydrous sodium sulfate (≥99.99 % trace metal basis, Merck, Darmstadt, Germany).



Instruments used in this work: high-purity germanium (HPGe) coaxial-type detector (model 7229P, Canberra Industries Inc., Meriden, CT, USA), high-purity germanium (HPGe) well-type detector (model GCW6023/S, Canberra Industries Inc., Meriden, CT, USA), a cold vapor atomic absorption spectrometer (model Hg-201 Semi-Automated Mercury Analyzer, Sanso

Seisakusho Co., Ltd., Tokyo, Japan), a liquid evaporative generator for oxidized mercury (Optoseven Ltd. & VTT Ltd., Espoo, Finland, amalgamation-atomic fluorescence spectrometer (model PSA 10.525 Sir Galahad, P S Analytical, Orpington, UK), a temperature-controlled bath (model R2, Grant Instruments Ltd., Cambridge, UK, circulator model GD20). Bell Jar elemental Hg calibrator (model PSA 10.555, P S Analytical, Orpington, UK), Cavkit elemental Hg vapor generator (model PSA 10.536, P S Analytical, Orpington, UK), capillary gas chromatography atomic fluorescence spectrometry (model PSA 10.725, P S

Analytical, Orpington, UK) with Agilent J&W, DB1, 15 m 0.53 mm ID, film thickness 1.50 µm.

## 2.2 Production of $^{197}$Hg radiotracer

Mercury labelled with radioactive $^{197}$Hg was used for all KCl sorbent trap experiments. Mercury enriched to 51.58 % in $^{196}$Hg isotope (only 0.15 % of $^{196}$Hg isotope is naturally present) was used to produce $^{197}$Hg ($t_{1/2}$ = 2.671 d) by a neutron capture reaction (n,γ). 2 mL of a 2 % HNO$_3$ ($v/v$) solution of enriched mercury solution was sealed into quartz ampoules. Quartz

ampoules were then irradiated in the central channel of the TRIGA Mark II (250 kW) reactor core channel (JSI, Ljubljana, Slovenia). High neutron flux (approximately $10^{13}$ cm$^{-2}$ s$^{-1}$ at thermal power of 250 kW) in the center of the reactor core caused the nuclear neutron capture reaction in solution to produce $^{197}$Hg during irradiation. Prior to irradiation, Hg concentration of the solution was determined by cold vapor atomic absorption spectrometry (CV AAS). The Hg concentration measured (93.3 µg mL$^{-1}$ of Hg) was the reference for Hg amounts that were used in all experiments. HgX$_2$ (X=Cl$^-$, Br$^-$) solutions and gases

were used in presented work; to clarify, all Hg related concentrations that will be presented in the manuscript will refer to Hg concentrations and not to HgX$_2$ concentrations if not explicitly stated otherwise. After irradiation, the Hg solution was transferred from the irradiated vial and diluted to appropriate Hg concentrations for subsequent experiments.

## 2.3 Determining $^{197}$Hg by using an HPGe detector

The activity of radiotraced Hg in solutions was measured by means of a well-type HPGe detector, while in gold traps and non-

liquid samples the activity was measured using a coaxial-type HPGe detector. All activity measurements were relative to standards obtained from the irradiated solution in each experimental run, considering the Hg concentration as described in the previous paragraph. The $^{197}$Hg activity was determined by evaluating γ-ray and X-ray emissions; experimental samples were measured in the same geometry as the standards. To obtain standards for well-type HPGe detector, triplicates of a Hg radiolabeled solution (8 mL, 2 % HNO$_3$ ($v/v$)) were transferred into glass vials. The standard solution was always diluted so

that the activity was similar to the activity of the measured sample. The $^{197}$Hg activity of the standards in the vials was measured using an HPGe well-type detector. Standards for the coaxial-type HPGe detector were obtained by $^{197}$Hg$^{2+}$ → $^{197}$Hg$^0$ reduction, performed in an impinger with a SnCl$_2$ solution (100 mL, 2 % SnCl$_2$ ($w/v$) and 0.5 % HCl ($v/v$)) which was purged for 10 min with N$_2$ carrier gas (purity 4.7, flowrate of 1 L min$^{-1}$). Purged $^{197}$Hg$^0$ was transferred to a gold trap by the carrier gas to obtain



a measurement standard. The absence of $Hg^0$ breakthrough was confirmed by placing an additional gold trap downstream the

main gold trap. Similar to liquid samples, standards for the coaxial-type gamma detector were made in triplicate. Any time

that new gold traps were prepared, new triplicate standards were also prepared following the same procedure. Gold traps were

prepared in quartz tubes (170 mm long, 6 mm inner diameter) by placing 15 mm in length of the absorbing material, which

was fixed in place by quartz wool. Absorbing material was prepared by dissolving 1 g of $HAuCl_4 \cdot xH_2O$ (gold chloride hydrate)

in 10 mL of Milli-Q water where 10 g of $Al_2O_3$ (corundum, 0.60 – 0.85 mm grain size) was added. The solution was then

evaporated in an automatic rotary evaporator and the remaining material was heated to 500 °C for 4 h in an argon atmosphere.

In order to reuse gold traps, they were heated to 300 °C for 30 s which released the bounded $^{197}Hg^0$. Complete release of $^{197}Hg$

was confirmed by evaluation of $^{197}Hg$ remains in the HPGe detector.

The evaluation of the characteristic γ-ray and X-ray emissions associated with $^{197}Hg$ decay (two doublet peaks: 67.0 + 68.8

keV and 77.3 + 78.1 keV) was made by computing peak areas Genie 2000 Gamma analysis software. All activities were

referred to a reference time by applying an equation derived from the exponential law of radioactive decay. Exact equations

that were used for the calculation of the activity and recovery are available in the supplementary material (Eq. S1 and S2)

(Ribeiro Guevara and Horvat, 2013; Koron et al., 2012; Ribeiro Guevara et al., 2007).

**2.4 Specificity of KCl sorbent traps, experimental design**

A scheme of the experimental setup for testing KCl sorbent trap specificity is shown in Fig. 1. Firstly, $Hg^{2+}$ (ranging from 0.1

to 1 ng) was reduced to $Hg^0$ in the impinger using $SnCl_2$ solution (100 mL, 2 % $SnCl_2$ (*w/v*) and 0.5 % HCl (*v/v*)). $Hg^0$ was

then purged out with $N_2$ carrier gas (flowrate of 1 L $min^{-1}$) for 10 minutes, passed through various types of KCl sorbent traps

(described below) and captured at the end by a gold trap.

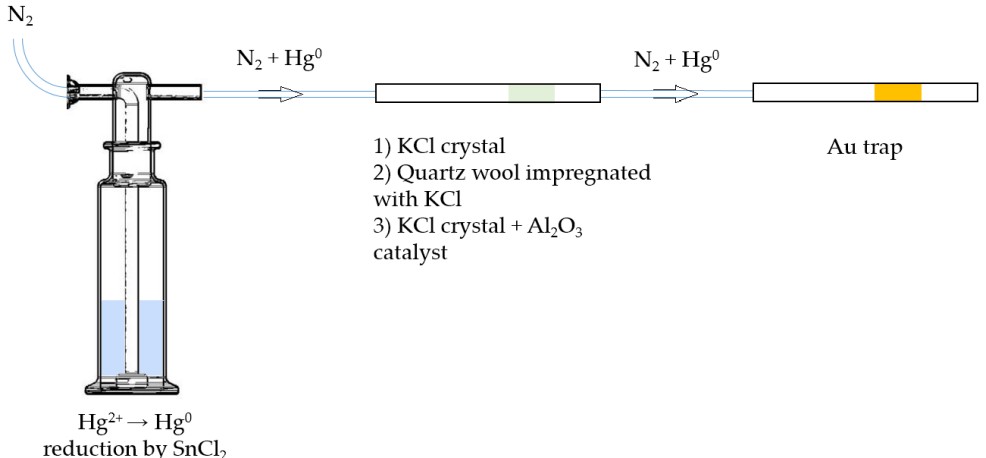

**Figure 1: Scheme of the experimental setup for testing the extent of undesirable $Hg^0$ retention on three different KCl sorbent trap**

**designs (not shown to scale).**



Three different types of KCl sorbent traps were used: KCl crystal, quartz wool impregnated with KCl and KCl crystal + $Al_2O_3$ catalyst. The latter was considered for an application that will be discussed in future work. Briefly, $Al_2O_3$ is intended to catalyze the reduction of $Hg^{2+}$ to $Hg^0$ in the next step of analysis (not discussed here). All sorbent traps were prepared in quartz tubes (170 mm long, 6 mm inner diameter). In design 1, KCl crystal was 15 mm long and fixed using quartz wool. In design 2, quartz wool impregnated with KCl was 70 mm long. Quartz wool impregnated with KCl was prepared by soaking quartz wool in 1 mol $L^{-1}$ KCl for 24 h, draining the excess solution and drying at 130 °C for 24 h. In design 3, KCl crystal was 5 mm long and $Al_2O_3$ catalyst part was 65 mm long. All three types of sorbent traps were tested new (no heating of the traps) and reused (traps heated to ≈ 600 °C three times prior to the experiment), resulting in six variations tested.

To determine the amount of $Hg^0$ collected on the KCl sorbent traps, they were leached with 20 mL of 10 % $HNO_3$ (*v/v*) + 5 % HCl (*v/v*) solution and $^{197}Hg$ in leachate was measured with a well-type detector. This acid mixture has previously been shown to completely leach Hg from KCl sorbent traps. Radiolabeled $Hg^0$ on gold traps was measured using a coaxial-type detector.

## 2.5 Stability of $Hg^{2+}$ loading on KCl sorbent traps, experimental design

In order to test $Hg^{2+}$ stability on KCl sorbent traps, the traps were exposed to ambient air flow for 30 min periods after loading with radiolabeled Hg. Potential formation of $Hg^0$ was captured downstream of the KCl sorbent trap by an Au trap. After each 30 min exposure period, the radiolabeled $Hg^0$ activity in the Au trap was measured in a coaxial-type detector. KCl sorbent traps were loaded with radiolabeled $Hg^{2+}$ again (to simulate the sampling process where new $Hg^{2+}$ is constantly adsorbed); the procedure was repeated 4 to 5 times. To assure that no $Hg^{2+}$ breakthrough occurred and that measured losses were only $Hg^0$, an additional KCl crystal trap was placed between the KCl sorbent trap and the Au trap to filter the potential $Hg^{2+}$ breakthrough. The $Hg^{2+}$ on the KCl trap filter was always below the detection limit of the gamma detector, so these results will not be shown in results and discussion section. Since this KCl trap filter was always free of Hg, it was later checked only from time to time for control. Figure 2 shows a diagram of all conditions studied in the stability tests.



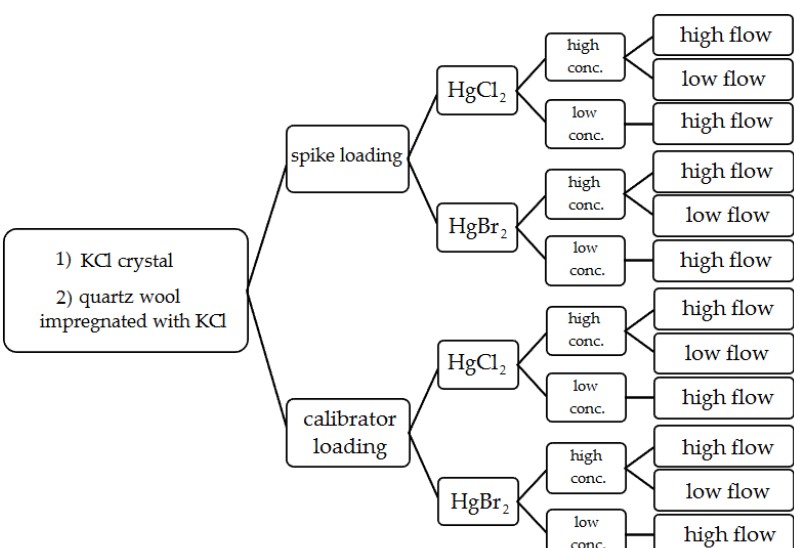

**Figure 2: Variations of performed experimental conditions for $Hg^{2+}$ stability on KCl sorbent traps during exposure in ambient air flow.**

Two types of $Hg^{2+}$ loadings were tested. The first type was a direct spike of $Hg^{2+}$ onto the sorbent traps. The second type of loading was done using an Optoseven evaporative gas calibrator. This instrument enabled $Hg^{2+}$ loading by evaporating $Hg^{2+}$ solution and injecting it into a carrier gas (Saxholm et al., 2020; Gačnik et al., 2021). The calibration gas was comprised of 0.07 mL min$^{-1}$ of $Hg^{2+}$ solution ($Hg^{2+}$ concentration depended on the concentration level tested) and 5 L min$^{-1}$ of carrier gas $N_2$. The calibration gas was formed in the evaporator at 125°C. The obtained calibration gas had a $Hg^{2+}$ concentration of 1178

ng m$^{-3}$ for high concentration tests and 5.90 ng m$^{-3}$ for low concentration tests. Previous work showed that the output of the calibrator is concentration dependent, these findings were taken into account when calculating the expected calibrator output (Gačnik et al., 2021). Two $Hg^{2+}$ species were tested for stability, $HgCl_2$ and $HgBr_2$. For $Hg^{2+}$ spikes, 4 % HCl ($v/v$) + 3 % $HNO_3$ ($v/v$) solution were used for $HgCl_2$, and 4 % HBr ($v/v$) + 3 % $HNO_3$ ($v/v$) solution were used for $HgBr_2$. Compounds and their concentrations were chosen based on the composition of NIST 3177 standard reference material (Mercuric Chloride

Standard Solution). In those cases where $Hg^{2+}$ was loaded using the Optoseven calibrator, a 0.1 % HCl ($v/v$) + 0.1 % $HNO_3$ ($v/v$) solution was used for $HgCl_2$, and 0.1 % HBr ($v/v$) + 0.1 % $HNO_3$ ($v/v$) for $HgBr_2$. In addition to two loading types and two Hg species, two different KCl sorbent trap materials were tested: KCl crystal and quartz wool impregnated with KCl. Each trap material was then tested under different experimental conditions: high concentration (> 50 ng)/low concentration (< 1 ng) and high air flow (400 mL min$^{-1}$)/low air flow (100 mL min$^{-1}$). All variations of experimental conditions, trap types, Hg

species and loading types were following Fig. 2 diagram.



## 2.6 Solubility of elemental Hg in KCl trapping solutions, experimental design

The solubility of elemental Hg in the 1 mol $L^{-1}$ KCl trapping solution was established using a method based on the Henry's law constant determination. From the dimensionless Henry's law constant (HLC), the amount of elemental Hg that would be collected in the KCl solution can be predicted. This approach was applied to a seawater matrix by Andersson et al (Andersson

et al., 2008).

The system consisted of an extractor stripper vessel composed of a jacketed borosilicate glass cylinder. This is shown schematically in Fig. 3. During the experiment, water at known temperature was pumped through the jacket from a temperature-controlled bath. The capacity of the extractor vessel was 1 L. At the bottom of the vessel there was an injection port connected to the argon carrier gas. The gas was bubbled inside the vessel through a glass frit on the bottom of the vessel.

The gas was produced from a mass flow meter at known flow. Two holes, one on the bottom and one on the top, allow the measurement of the temperature of both the solvent and the headspace. At the top of the vessel, the pressure was measured using an absolute pressure meter.

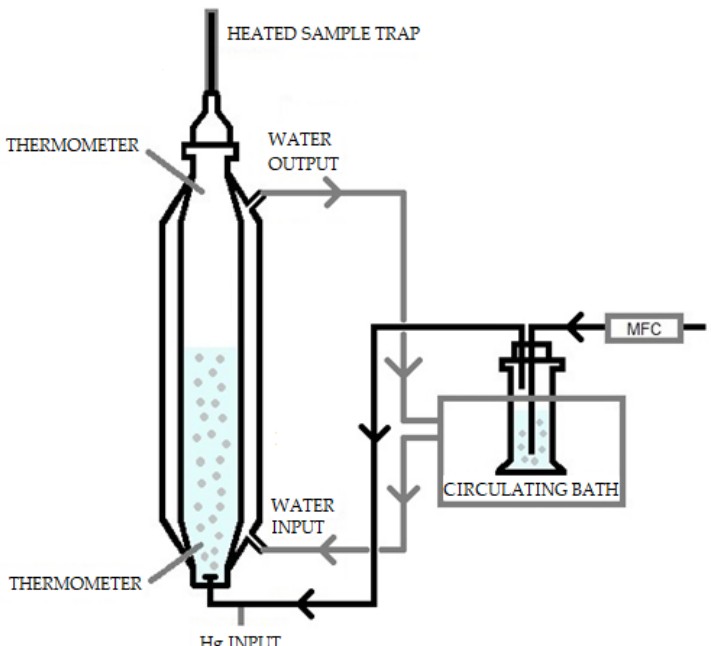

**Figure 3: Henry's law extractor stripping vessel arrangement.**

The mercury which is purged from the vessel was collected using a heated gold trap. Before every experiment, the recovery of the gold traps was checked to ensure that mercury collection was quantitative. The recovery was also checked after each experiment. The mercury extracted was compared to the mass of Hg injected as an indication of the success of the experiment and absence of oxidized mercury which would otherwise be trapped in solution. The KCl also included 11 mL $L^{-1}$ of reductant solution (20 % (*w/v*) SnCl$_2$ in 10 % (*v/v*) HCl), used to avoid oxidation.



Elemental mercury ($Hg^0$) vapors were injected from a bell-jar into the gas flow of the extractor vessel. The bell-jar had a thermometer where the temperature was checked before the gas was pulled via a gas tight syringe. The temperature and volume of gas are known so the mass of mercury injected could be calculated using the Dumarey equation (Dumarey et al., 2010). To make a correction on the measured mass flow and calculate the real flow the vapor pressure must be known. The vapor pressure of 1 mol $L^{-1}$ of KCl at 5°C was assumed to the same as water (Lide, 2007). The recovery of the spiked mercury into the

extraction vessel was checked after the experiments. The recovery is a good indication of the success of the experiment and of the absence of oxidized mercury.

The gas coming out from the vessel was collected using heated gold traps for fixed intervals of time. These gold traps were then analyzed to determine the mass of Hg released during each time interval. For this experiment the volume of KCl in the vessel, the pressure, the measured gas flow rate, the exact interval of each extraction time and the temperature must be known

exactly. The calculation of dimensionless HLC was done according to Andersson et al. (Andersson et al., 2008) and is shown in supplementary material. The value of the gold trap blank was subtracted from the mass of Hg measured for every extraction. This improved the linearity of the data giving a better measurement of the dimensionless HLC. Usually, the first point of the extraction deviated slightly from linearity because it included the signal rise time from the point of Hg vapor injection. During the signal rise period the test solution is not under equilibrium and therefore the first point was ignored. Additionally, the final

extraction points if considered to be close to the quantification limit are ignored as the uncertainty is higher which can affect the linear regression.

## 2.7 Oxidation of elemental Hg in KCl trapping solutions, experimental design

The experimental arrangement for studying the selectivity of the KCl trapping solution is shown in Fig. 4. A PSA 10.536 elemental Hg generator was used to generate a continuous stream of calibration gas of known concentration (nominally

20 µg $m^{-3}$) and flowrate. This is considerably higher concentration of elemental Hg than ambient air concentration as the original scope of this work was to study the Hg concentration range in flue gases. The total flow generated was 5 L $min^{-1}$ and both nitrogen and air were studied. A slipstream flow of 0.5 L $min^{-1}$ was pulled through the test impingers using a vacuum pump and mass flow controller arrangement. An impinger solution of 100 mL of 1 mol $L^{-1}$ KCl was used as the test solution maintained at 5°C in a water chiller bath. An empty impinger and an impinger of silica gel were also included to ensure that

the gas going to the mass flow controller was dry. All tests were conducted over a 2-hour period during which an absolute mass of mercury of 1200 ng was passed through the impinger train. The KCl trapping solution was then analyzed by GC-AFS with aqueous phase propylation liquid-liquid extraction to determine the oxidized Hg.

The same apparatus was used to test the stability of oxidized Hg in the KCl trapping solution. In this case a known mass of $HgCl_2$ was added to the KCl solution. The apparatus was then run using the same conditions as described above but without

elemental Hg being introduced. In this experiment the concentration of oxidized Hg should remain the same after the 2-hour period of sampling. A concentration decrease would be indicative of $HgCl_2$ being reduced. After running the test, the KCl



trapping solution was then analyzed by GC-AFS with aqueous phase propylation liquid-liquid extraction to determine the oxidized Hg.

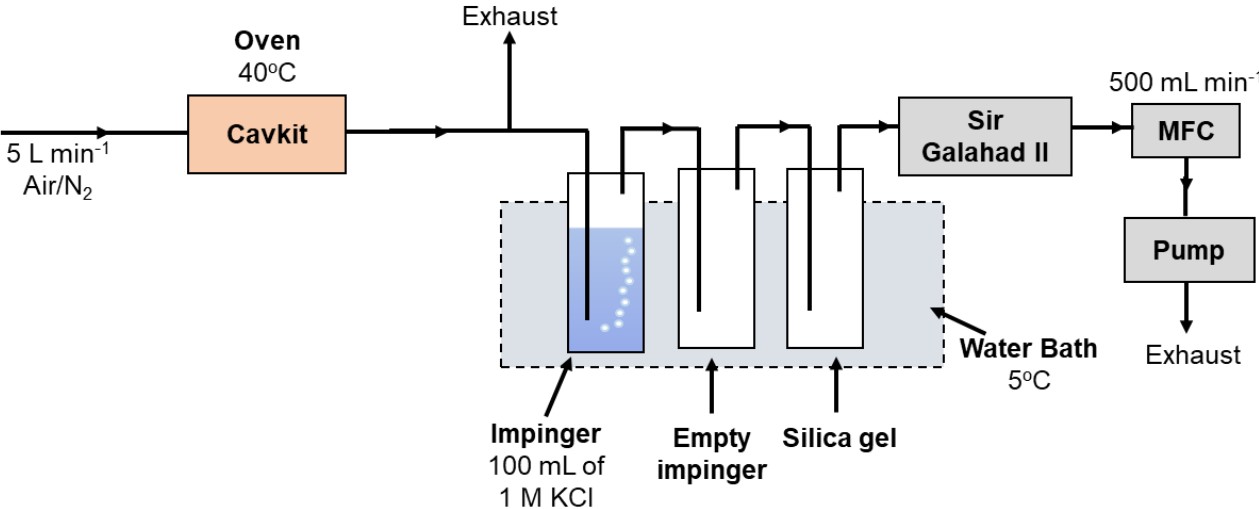

**Figure 4: Experimental arrangement for KCl trapping experiments.**

**2.8 Determination of oxidized Hg in KCl trapping solution, experimental design**

The KCl solution was propylated in the presence of an acetic acid-acetate buffer. This converts $Hg^{2+}$ to dipropylmercury. The derivatized mercury was then transferred and concentrated in an organic phase for injection into the GC-AFS instrument. To 100 mL sample, 5 mL buffer (0.5 M acetic acid) was added. The pH was then adjusted to 3.9. To this solution 500 µL of 2,2,4-

trimethylpentane and 1 mL of the alkylation reagent were added. The solution was then shaken vigorously for 10 min and the trimethylpentane phase transferred to a GC vial. The sample was dried over anhydrous sodium sulfate and then analyzed by GC-AFS. Calibration was achieved by preparing mixed organometallic and $Hg^{2+}$ standards and blank, subjecting them to the same sample preparation. An injection volume of 2 µl was used in the splitless mode of operation with the injector at 250°C. The GC temperature program used is summarized in the supplementary material. The eluent coming out from the column was

thermally treated at 800°C to breakdown organomercury compounds to elemental Hg before introduction to the AFS detector.

**3. Results and discussion**

**3.1 Specificity of KCl sorbent traps**

The aim was to determine the extent of unwanted retention of $Hg^0$ on KCl sorbent traps. This was achieved by transferring a known amount of $Hg^0$ in the carrier gas through a $Hg^{2+}$ specific sorbent trap (where $Hg^0$ retention is unwanted). The results of

the experiment are shown in the Table 1. The column "retained $Hg^0$" represents $Hg^0$ retained on the KCl sorbent trap. The column "mass balance" represents the sum of $Hg^0$ on the KCl sorbent trap and $Hg^0$ on a gold trap. All values are shown as



percentages relative to the Hg amount that was purged through the system. As already mentioned, $Hg^{2+}$ breakthrough was never present since it was negliglible.

**Table 1: $Hg^0$ retention on various KCl sorbent trap designs, comparison of new and reused designs. Results are presented in**
**percentage of the initially purged $Hg^0$ amount. Less than 1 ng of Hg was used.**

| Trap description | New / reused | Retained $Hg^0$ [%] | Mass balance [%] |
|---|---|---|---|
| KCl crystal + $Al_2O_3$ catalyst | New | 0.00 | 99.1 |
| | | 0.00 | 96.7 |
| | Reused | 11.5 | 102 |
| | | 18.0 | 101 |
| | | 23.8 | 101 |
| | | 9.46 | 101 |
| KCl crystal | New | 0.13 | 95.9 |
| | | 0.14 | 102 |
| | | 0.20 | 96.2 |
| | Reused | 4.10 | 95.4 |
| | | 7.12 | 92.7 |
| | | 2.41 | 94.5 |
| Quartz wool impregnated with KCl | New | 0.05 | 98.2 |
| | | 0.23 | 109 |
| | | 0.23 | 93.3 |
| | Reused | 0.35 | 100 |
| | | 0.10 | 101 |
| | | 0.64 | 100 |

It can be seen that trap designs containing KCl crystals (KCl crystal + $Al_2O_3$ catalyst and KCl crystal traps) retained much more $Hg^0$ when reused which is not desirable; $Hg^{2+}$ specificity is required. The amount retained also varied greatly for reused sorbent traps. Since the KCl has a melting point of 770°C, this could mean that the morphology of KCl has changed at
experimental temperatures ($\approx$ 600°C) approaching the melting point. The morphological change could potentially explain the increased $Hg^0$ binding when using reused KCl sorbent traps. Traps that were unused prior to the experiment ("new" traps) always retained very small amounts of $Hg^0$ (< 0.3 % for all designs). Mass balances were quantitative in all cases; therefore, these results can be trusted with a high level of confidence. From these findings, it can be concluded that new traps perform better than reused traps and therefore, to prevent the formation of artifacts only new traps should be used. It is not necessary
to correct the measured values for the obtained recovery as losses are regularly lower than the variability of the recovery





(standard deviation up to 4 %, losses up to 0.2 %). These results on the specificity of KCl sorbent traps could also be useful

for KCl-coated denuders, as they are analyzed in multiple heating cycles of 700°C and are reused regularly. Although the values may differ from this work, the effect of reuse on the specificity of denuders could be similar. Because denuders are used for GOM sampling upstream of the Au traps for GEM sampling, this could result in GEM retention on KCl-coated
denuders that are intended to be GOM-specific.

## 3.2 Stability of $Hg^{2+}$ loading on KCl sorbent traps

Exposure of the loaded sorbent trap to air flow changed the geometry of the radiolabeled $Hg^{2+}$ loading. This affected the measurement as shown in Fig. 5, resulting in biased results. This effect was presumed based on the fact that mass balance was always considerably above 100 %. The figure illustratively shows a sorbent trap that is placed over a coaxial-type gamma
detector before and after exposure to air flow. The air flow shifted the distribution of the radiolabeled $Hg^{2+}$ along the trap. Prolonged exposure to the air flow caused greater shifts. Due to geometrical effects, the detection system has higher detection efficiency for a radioactive source on the axis of the detector than a source at the same distance from the detector surface but far from the axis. Therefore, a distribution of the radiolabeled Hg extended to the axis of the detector generates a higher recording on the detection systems than a distribution compacted to the detector border, with the same activity. Measurement
after exposure to the air flow would therefore result in greater apparent sample activity than measurement prior to exposure to air flow. Due to this observation, it was not possible to verify the overall mass balance. On the other hand, this did not affect the measurement of Hg losses ($Hg^0$) captured on Au trap, as Hg forms a strong amalgam with Au. Therefore, only the measured losses are presented in the tables below.

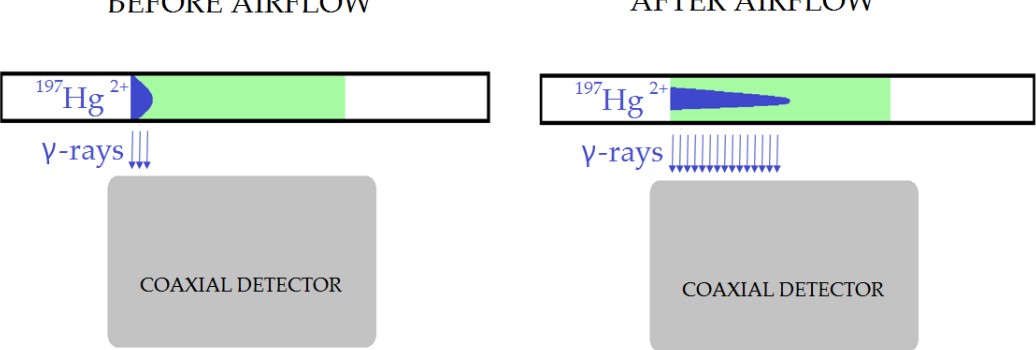

**Figure 5: Effect of radiotracer distribution change along the trap on the activity measurement after exposure of KCl sorbent traps to air flow (not shown to scale).**

The results of the stability tests are shown in the figures below (Fig. 6, Fig. 7, Fig. 8 and Fig. 9). Full results are available in supplementary material (Table S2, Table S3, Table S4 and Table S5). Losses are presented as a percentage relative to the cumulative amount of Hg that was spiked up to that time period. Some results have four time periods (marked with an asterisk
in the figures) while others have five time periods due to time constraints of each measurement day.

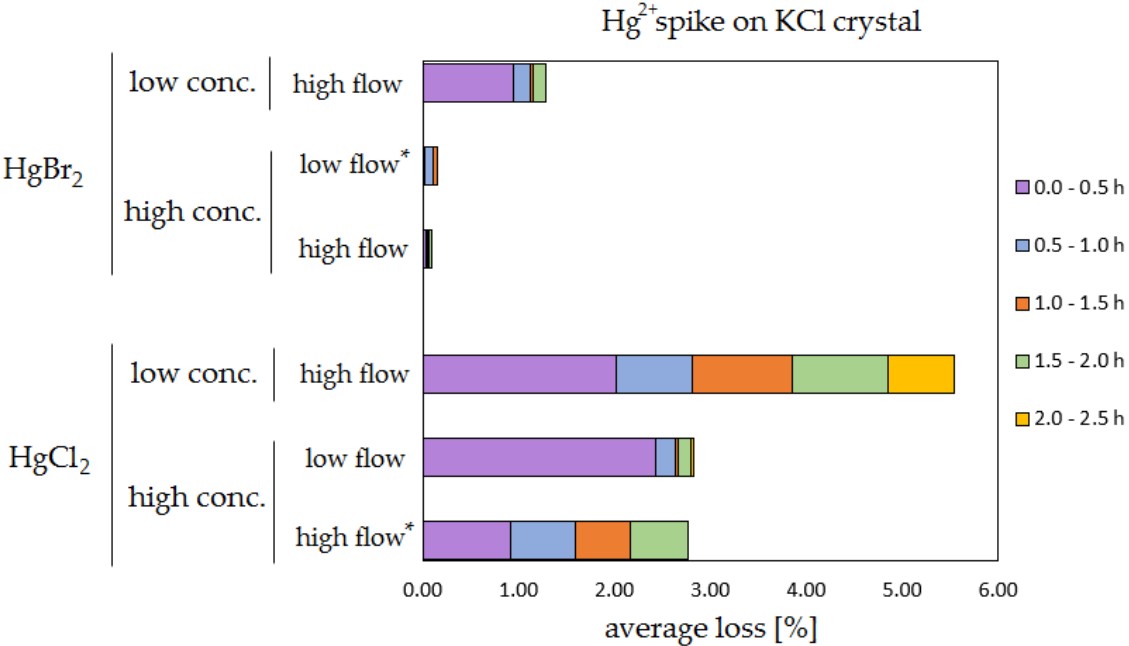

**Figure 6: Results of the stability test for the $^{197}Hg^{2+}$ (radiotracer) spike on KCl crystal. Low concentrations were loaded with less than 1 ng Hg per time period and high concentrations were loaded with more than 50 ng of Hg per time period. Low air flow experiments were performed with 100 mL min$^{-1}$ air flow while high air flow experiments were performed with 400 mL min$^{-1}$ flow.**
**Asterisk marks the results that only have the first 4 time periods.**

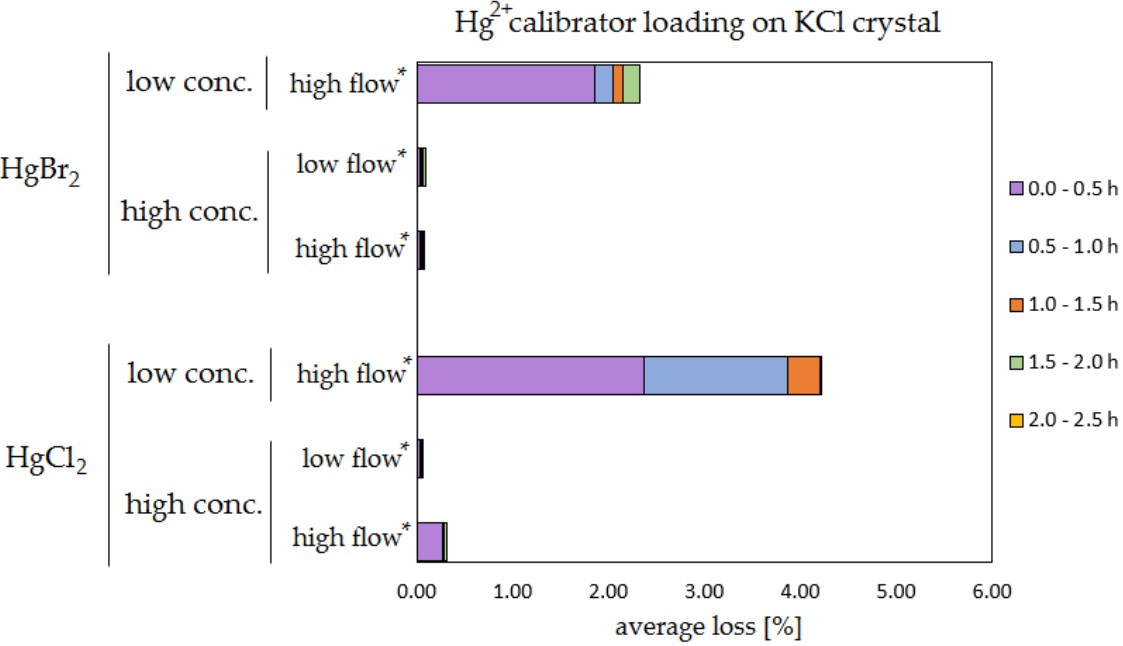





**Figure 7: Results of the stability test for the calibrator loading of $^{197}Hg^{2+}$ (radiotracer) on KCl crystal. Low concentrations were loaded with less than 1 ng Hg per time period and high concentrations were loaded with more than 50 ng of Hg per time period. Low air flow experiments were performed with 100 mL min$^{-1}$ air flow while high air flow experiments were performed with 400 mL min$^{-1}$ flow. Asterisk marks the results that only have the first 4 time periods.**

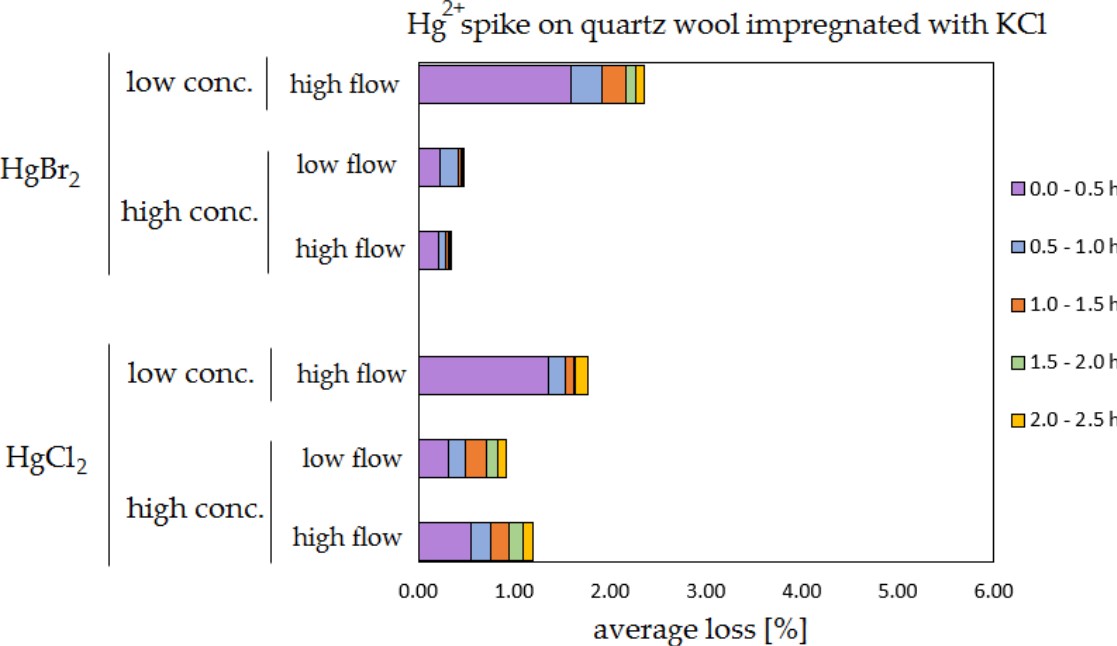

**Figure 8: Results of the stability test for the $^{197}Hg^{2+}$ (radiotracer) spike on quartz wool impregnated with KCl. Low concentrations were loaded with less than 1 ng Hg per time period and high concentrations were loaded with more than 50 ng of Hg per time period. Low air flow experiments were performed with 100 mL min$^{-1}$ air flow while high air flow experiments were performed with 400 mL min$^{-1}$ flow.**



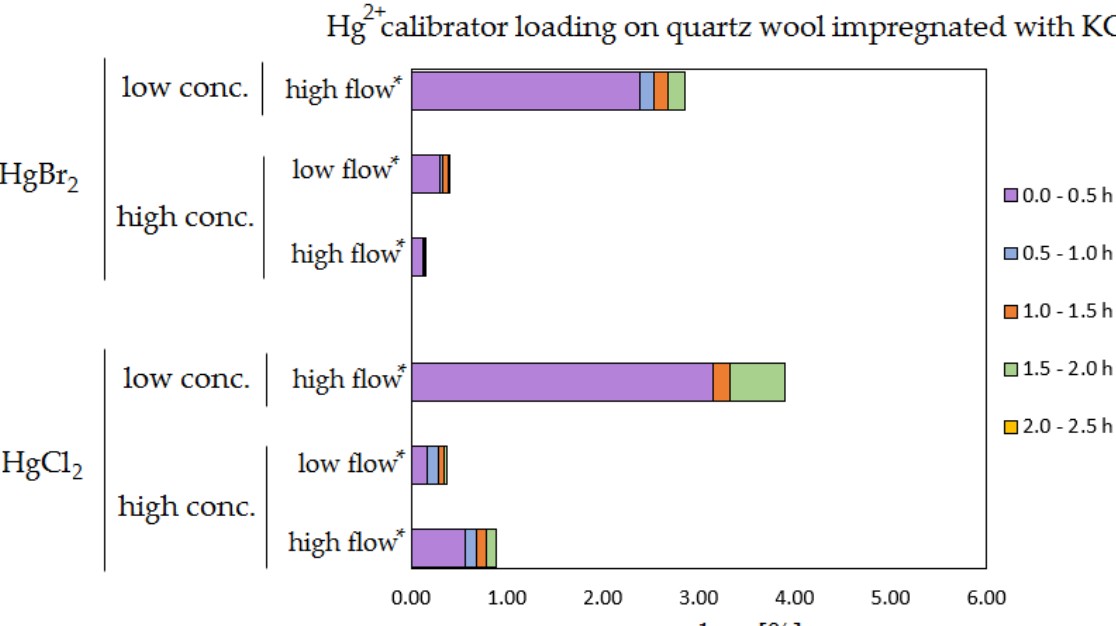

**Figure 9: Results of the stability test for the calibrator loading of $^{197}$Hg$^{2+}$ (radiotracer) on quartz wool impregnated with KCl. Low concentrations were loaded with less than 1 ng Hg per time period and high concentrations were loaded with more than 50 ng of Hg per time period. Low air flow experiments were performed with 100 mL min$^{-1}$ air flow while high air flow experiments were performed with 400 mL min$^{-1}$ flow. Asterisk marks the results that only have the first 4 time periods.**

The clear trend observed in the tables presented was that the relative losses were in almost all cases higher in the low concentration experiments (5.54 % max. losses) than in high concentration experiments (2.79 % max. losses, under 1 % in most cases). Nevertheless, the absolute losses (tables and graphs above are presented in relative values) were still higher in the high concentration experiments than in the low concentration experiments. In addition, the first interval (0 – 0.5 h) had statistically significant maximum relative losses during the whole stability test (Kruskal–Wallis test, p < 0.001; pairwise multiple comparison procedures (Dunn's Method), p < 0.05 for 0 – 0.5 h period against other periods). Because the variation in low/high air flow did not cause significant differences in the overall Hg$^{2+}$ losses (paired t-test, p = 0.471), the low air flow tests were omitted in the low concentration stability tests. The HgCl$_2$/HgBr$_2$ and calibrator/spike loading variations did not also cause any significant differences in Hg$^{2+}$ losses during the stability tests.

Longer sampling times are often used for low concentrations of Hg$^{2+}$ (the amount of Hg$^{2+}$ collected from the ambient atmospheric samples is in the order of picograms), so the losses as observed in the above experiments should be taken into account when evaluating atmospheric Hg speciation measurements. In Hg speciation measurements, reduction of Hg$^{2+}$ to Hg$^0$ during sampling may result in a positive bias for Hg$^0$ (gaseous elemental mercury, GEM) and negative bias for Hg$^{2+}$ (gaseous oxidized mercury, GOM) measurement. The considerations mentioned above should be considered carefully, especially when longer sampling times are required.





### 3.3 Solubility of elemental Hg in KCl trapping solutions using dimensionless Henry's law constant

Using the experimental data taken from one of the tests using 1 mol L$^{-1}$ KCl at 5°C, the mass of Hg released during each two-minute interval is shown graphically in Fig. 10. The natural logarithm of the mass of the Hg released during each interval was

plotted to provide the α term using the slope, see Fig. 11. The slope was established using a linear regression weighting the errors in $\ln\left(m_{Hg}(n)\right)$ using the calculated expanded uncertainty with a coverage factor $k=2$.

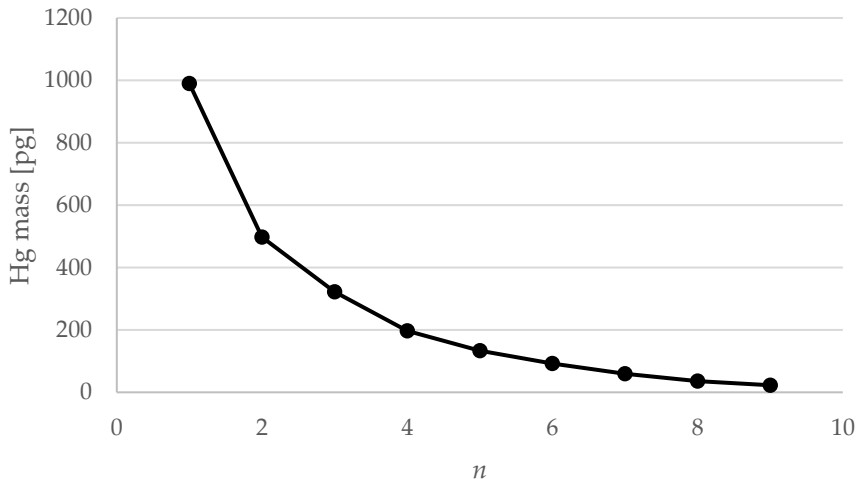

**Figure 10: Example $m_{(Hg)}$ against extraction number ($n$) for 1 mol L$^{-1}$ KCl at 5°C.**

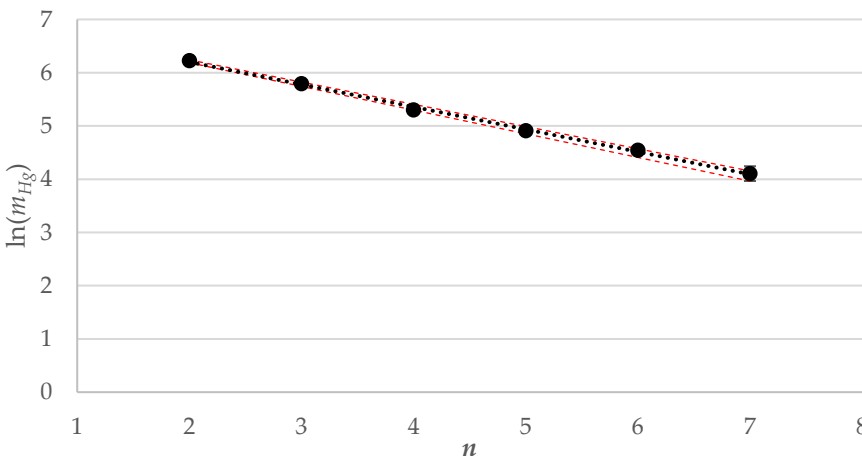

**Figure 11: Example ln($m_{Hg}$) against extraction number ($n$), linear regression for KCl at 5°C (red error bars ± 1 $u_c$ ($k$ = 1)).**

As can be seen from Fig. 11, plotting the logarithm of the Hg mass extracted against the extraction number gives a linear relationship that allows the calculation of the dimensionless HLC. A combined dimensionless HLC was found to be 0.1713 with an expanded uncertainty ($k$=2) of 0.0093 (calculation and derivation of needed equations is located in the supplementary



material, Table S1). The dimensionless HLC can be used to calculate the elemental mercury concentration collected in the KCl
solution at an equilibrium condition.

### 3.4 Oxidation of elemental Hg in KCl trapping solution

The oxidation of elemental mercury found for the nitrogen and air matrices was 2.9 % and 3.0 % respectively. When the KCl
trapping blank solution was tested it was found to be below the GC-AFS method detection limit (1 pg). Impurities present in
the KCl trapping solution appear to oxidize a small percentage of the elemental Hg vapor which was continuously introduced
during the test. As very little difference was found when comparing air and nitrogen it is reasonable to assume that this
oxidation was not due to aerial oxidation. The results show that the KCl solution will also collect a small quantity of elemental
Hg due to its solubility and this elemental Hg was observed in the chromatograms as shown in the Fig. 12. This was not
observed in standard solutions or blanks. The elemental Hg response was not quantified because the procedure used is not
considered quantitative for elemental Hg as this species does not undergo derivatization.


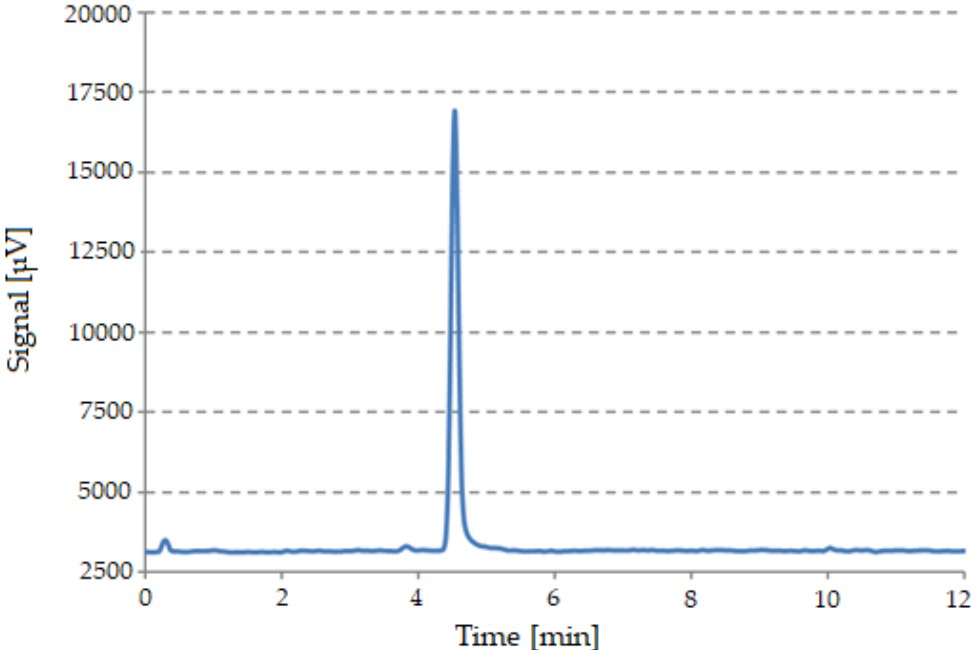

**Figure 12: Example chromatogram for aqueous phase propylation GC-AFS showing GEM oxidation in KCl trapping solution.
Oxidized Hg observed as dipropylmercury (retention time of 4.6 min) and elemental Hg (retention time of 0.3 min)**



## 3.5 Retention of oxidized Hg in KCl trapping solution

In this test bubbling the KCl solution spiked with $HgCl_2$ with nitrogen or air for the 2-hour sampling period gave a recovery of 102.3 % and 101.8 % respectively indicating that no $HgCl_2$ was lost from the KCl solution. The slight increase is within the measurement uncertainty of the propylation-GC-AFS.

## 3.6 Predicted bias calculation using KCl trapping solution at atmospheric Hg speciation concentrations

The experimental data indicates that GEM oxidation (section 3.4) and GEM solubilization (section 3.3) will occur in KCl
trapping solutions. The combination of these two factors will produce a positive bias which can be predicted using Eq. 1.

$$Predicted\ GOM\ bias\ (\%) = \frac{GEM_{oxid} + GEM_{sol}}{GOM_{native}} \times 100\% \qquad (1)$$

For example, if a 1 $m^3$ ambient air sample containing 2 ng $m^{-3}$ GEM and 0.002 ng $m^{-3}$ GOM ($GOM_{native}$) was sampled through a 1000 mL KCl solution the bias can be predicted as follows: By applying the dimensionless HLC obtained from the presented work and using the equations available in supplementary material, the calculated contribution due to solubility ($GEM_{sol}$) will
be 0.012 ng $m^{-3}$. Oxidation of 3 % of GEM was found in the KCl trapping solution. This equates to $GEM_{oxid}$ of 0.060 ng $m^{-3}$ for this example. The apparent gas concentration in the KCl solution would be the summation of soluble GEM (0.012 ng $m^{-3}$), oxidized GEM (0.060 ng $m^{-3}$) and the native GOM (0.002 ng $m^{-3}$) equating to 0.074 ng $m^{-3}$ rather than the expected 0.002 ng $m^{-3}$. From this simple calculation using Eq. 1 a GOM bias of 3500% can be predicted due the solubility and oxidation of GEM in the KCl solution.

## 4 Conclusions

KCl sorbent traps show good stability for most of the experimental conditions tested in this work. When KCl sorbent traps are used for ambient Hg concentration, the extent of GOM losses can reach up to 5.5 % (2 % to 3 % on average). When calculating the overall uncertainty of the atmospheric Hg speciation, these losses should be taken into account as the sampling uncertainty. Reuse of KCl sorbent traps resulted in large reduction in their specificity. Because the reuse of KCl-coated denuders (for
ambient GOM sampling) is also common practice, a similar reduction in specificity could also occur for KCl-coated denuders. As denuders are the most commonly used means of sampling GOM, this poses a potential challenge which should be addressed. The [197]Hg radiotracer proved to be a suitable tool for studying the metrology and processes occurring during atmospheric mercury speciation. In the future, the [197]Hg radiotracer could be applied for verifying other GOM sampling methods such as denuders and different membrane filters.
KCl trapping solutions cannot be considered truly selective for GOM measurements for mercury atmospheric speciation measurements. GEM will be captured in the solution due to both oxidation and solubilization producing a large bias in the GOM measurement.

**Data availability**

Data is contained within the article or supplementary material. Other data used in this study can be acquired upon request to the corresponding author.

**Author contributions**

Conceptualization, JG, SRG, JK, WTC, and MH; Funding acquisition, MH; Methodology, JG, IŽ, GDF, MAD, SRG; Project administration, MH; Supervision, JK, WTC and MH; Validation, MAD, JG and IŽ; Writing—original draft, JG and WTC; Writing—review & editing, JG, IŽ, MAD and MH.

**Competing interests**

The authors declare that they have no known competing financial interests or personal relationships to influence the work reported in this paper.

**Acknowledgements**

Financial support from the project Integrated Global Observing Systems for Persistent Pollutants (IGOSP) funded by the European Commission in the framework of program "The European network for observing our changing planet (ERA-PLANET)", Grant Agreement: 689443 is also acknowledged. The authors would like to thank the TRIGA reactor staff at the Reactor Infrastructure Centre of the JSI for their availability and cooperation at all times. We would also like to thank Jarkko Makkonen and Timo Rajamäki for supplying us with the calibrator and for operational advice. The authors would like to thank Ingvar Wängberg at IVL Sweden for initial guidance and providing detailed information regarding the extractor vessel apparatus.

**Financial support**

This research has been supported by: project 16ENV01 MercOx which has received funding from the EMPIR program co-financed by the Participating States and from the European Union's Horizon 2020 research and innovation program; Slovenian Research Agency (ARRS), grant number P1-0143 and PR-52044; Metrology Institute of the Republic of Slovenia (MIRS), contract number C3212-10-000071 (6401-5/2009/27).

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
