# Peer review of "Behaviour of KCl sorbent traps and KCl trapping solutions used for atmospheric mercury speciation: stability and specificity"

_Atmospheric Measurement Techniques, 2021_

## Author Response (AR1)

**RC1** response**

Dear reviewer,

thank you for the comments. Your comments are marked italic.

Abstract-the authors need to note that GOM and PBM are formed in the atmosphere not just emitted from sources

We have changed the sentence in line 13 which now goes as following: "GOM and PBM can also be formed in the atmosphere; their sampling is the most problematic step in the analytical procedure."

*I disagree that the work done at ambient air concentrations are limited. The cation exchange membrane method detection limits are well below ambient concentration*

We are aware of the work with cation-exchange membranes, though we were trying to stress that the ambient concentration work on KCl sorbent traps and KCl trapping solutions alone was limited. Ambient concentration work with other sampling methods (i.e. cation exchange membranes) is of course available in the literature. We rephrased this in line 15: "GOM sampling with speciation traps composed of KCl sorbent materials and KCl trapping solutions are commonly used sampling methods, although the work done with them at ambient air concentrations is limited."

*Line 26- first sentence of the abstract need a reference. Second sentence this is simply not true!*

Reference for the first sentence of introduction was added and the second sentence was removed (lines 26-28).

**Line 40 -PTFE membranes are used to collect PBM not GOM. Nylon and CEM collect RM when there is no PTFE in front.**

Line 40 was corrected according to your suggestions, the sentences were rearranged in lines 38-42 to: "For gaseous oxidized mercury (GOM) the main sampling and preconcentration methods are: KCl-coated denuders (Bu et al., 2018), KCl impinging solutions (impingers, adaptations of the Ontario Hydro method (ASTM International, 2016)) and KCl sorbent traps (U.S. Environmental Protection Agency, 2017; Prestbo and Bloom, 1995). Cation-exchange (CEM) or nylon membranes collect reactive mercury (RM – sum of GOM and PBM) but can also be used for GOM sampling if poly(tetrafluoroethylene) (PTFE) membranes (PBM collection) are placed upstream of them (Bu et al., 2018; Huang et al., 2013; Gustin et al., 2021)

Line 46 start new paragraph with "Most".

New paragraph was added as suggested.

*Line* 65- *Again, it has been well documented that CEM can be used to measure ambient concentrations*

In line 65 we just excluded the statement about limitation for ambient concentration: "CEM and polyethersulfone membrane (PES) have been shown to be the most quantitative sorbents."

**Line 69- should be that CEM and nylon retain GOM if placed downstream of a PTFE membrane.**

Line 69 was corrected according to your suggestions: "Authors suggest that PTFE membranes retain mostly PBM while CEM and nylon membranes retain RM without an upstream placed PTFE membrane or GOM with an upstream placed PTFE membrane (Gustin et al., 2021)."

I know this may be a bit unconventional, but to make the results clearer would it be better to combine the methods and results. For example, section 3.1 List the aim of the experiment, describe the experiment, discuss the results. I think this might make this paper easier to follow.

We think that having only one section instead of separated Methods and R&D would make sense for some sections but for many it would make things less clear and less readable. Therefore, we decided to keep the article in this format, since it also fulfills the AMT's guidelines for authors.

Section 3.4 would be good to add a description of the other air used including any air chemistry you might have.

The air was obtained from our inhouse air compressor system and is classified to ISO 8573-1:2010. We have added the reference into the text in line 232. Other air was not tested.

Line 372-Please note recent data with membranes and dual-channel systems have demonstrated that GOM is typically 25% in ambient air.

We have added a calculation for a higher GOM percentage and explained that the bias is dependent on the GEM:GOM ratio in the ambient sample: "The bias depends on the GEM:GOM ratio, the higher the percentage of GOM relative to GEM, the lower the bias will be. For example, a similar calculation as above but with 1.980 ng m-3 GEM and 0.02 ng m-3 GOM results in 456 % bias instead of 3500 %."

**Conclusions-It is important to note that most people do not now use the 1130/1135 unit on the Tekran system and papers with these data are not even being sent out for review.**

I am struggling to find the connection between what you stated and our work. Additionally, there are still papers published in the literature that use data from 1130/1135 unit on the Tekran system. Examples: Slemr et al. 2020 (Atmospheric Chemistry and Physics), Wang et al. 2021 (Atmospheric Research), Mason et al. 2021 (Atmospheric Environment), Griggs et al. 2020 (Atmosphere).

**Figures 10 and 11 remove gridlines. You might consider putting figures like this in the Supplemental Information.**

We have removed the gridlines as suggested.

**RC2** response**

Dear reviewer,

thank you for the comments. Your comments are marked italic.

Line 16-18: the stability test of Hg(II) showed a highest loss of 5.5% (mostly around 1-2%), this is overall small relative to the analytical uncertainty of atmospheric GOM (could be biased by several times). I do no think a correction of such a small loss would improve the analysis of atmospheric GOM. I agree that a much lower loading of Hg(II) (e.g., 5-100 pg) may caused higher loss of Hg(II), but this was not done in this study. Therefore, based on the finding of this study, it is no practicable to draw a conclusion the GOM measurements should be corrected by the loss of Hg(II) (e.g., 5%).

I agree with the comment, there are biases of much higher orders than 5% for GOM. We have therefore excluded the sentence "GOM losses should be taken into account when using KCl sorbent traps for atmospheric Hg speciation, especially at low ambient GOM concentrations."

Line 156: why a KCl solution with a concentration of 1 mol L-1 was used to soak the quartz wool? Have the authors tried other concentrations? Would a higher concentration improve the sampling of GOM or cause a higher retention of GEM?

We have only tried 1 mol L-1 concentration. We have tested a variety of other conditions in the stability test (trap type, loading type, species loaded, high/low airflow, high/low species concentration). Solubility of KCl is the delimiting factor for increasing the concentration, highest being 4.5 mol L-1 at 20 °C. This would not be a major increase from 1 mol L-1 therefore we think that it would not change the outcome of the experiment.

Line 157-158: these KCl sorbent traps were reused by heating to a temperature of 600 °C. This temperature is relatively higher than the traditional temperature setting (e.g., 500-550°) for the desorption of GOM from KCl coated denuders. I think this may change the morphology of KCl, as suggested by the authors.

Though 500-550 °C is used for denuders, KCl sorbent traps are mostly heated at temperatures around 700 °C i.e. Lumex speciation traps and their pyrolysis with AAS detection. In literature, there are instances of even higher temperatures being used.

Line 159-160: the authors used a mixed acid solution to leach the Hg(0) collected on the KCl sorbent traps, I think these treatments would also change the chemical properties of KCl traps, which may increase the retention of GEM by the sorbent traps.

This is a misunderstanding: once traps were leached with acid, they were not reused again. They were "re-used" before the start of experiment by heating to  $\approx 600$  °C three times prior to the experiment (as described in line 158).

Line 182-183: The authors should explain why these two experiments would precisely generate HgCl2 or HgBr2. Have the authors determined the Hg(II) compounds in these two solutions?

The concentration of HgCl2 or HgBr2 compounds was measured by CV-AAS as described in section 2.2. We were aware of the exact composition of  $Hg_xCl_x$  and  $Hg_xBr_x$  species in the

solution by calculating the species abundance from the equilibrium constants. We have added a reference to the exact calculation which is described in our previous work. The added sentence goes as following: "By equilibrium calculations described in the work of Gačnik et al. we confirmed that the spiking solutions contained only HgCl2 and HgBr2 without other HgxClx or HgxBrx species (Gačnik et al., 2021)".

Line 276-280: In my opinion, an over interpretation of the results is not of significant scientific values. As mentioned above, the leaching and reuse processes of the KCl sorbent traps would probably change the morphology of KCl traps (especially for the KCl crystal and KCl crystal +Al2O3 catalyst), and they are quite different from the processes for the reuse of KCl denuders.

We have excluded this interpretation to your suggestion.

*Line 324-325:* A comparison of the absolute losses between the low and high loading of Hg(II) is meaningless.

We have excluded this comparison to your suggestion.

Line 331-336: I agree with the authors that loss of Hg(II) could occur during the sampling. However, the losses could be associated with many factors, such as quantity of Hg(II) on the sorbent traps, sampling and flushing flow rate, ozone, humidity, etc.. In this study, the loading of Hg(II) is much higher than the real atmospheric conditions, and the air flow was also lower than the GOM sampling flow rate (e.g., 10 LPM) and flushing flow rate during desorption processes (e.g., 1.0 LPM). Therefore, using the loss rate determined from this study to correct GOM measurement is not expected.

We changed the paragraph to the following: "Longer sampling times are often used for low concentrations of Hg2+ (the amount of Hg2+ collected from the ambient atmospheric samples is in the order of picograms), therefore some losses of GOM will be observed most of the time. Losses depend not only on the parameters tested in our work, but also on meteorological conditions (e. g. humidity, presence of ozone, temperature...)."

Section 3.6: A GOM bias of 3500% for the using the KCl solution trap is extremely higher. The authors should specify that the KCl solution trap should be not relevant for field GOM sampling. Note that KCl solution trap is generally used for the sampling of GOM in flue gas.

We have added a sentence to your suggestion: "The calculated biases show that KCl trapping solutions are not appropriate for ambient GOM sampling, while they are still a valid choice for flue gas sampling (high GOM concentrations)."

The fact that these solutions are generally used for sampling of GOM in flue gas is already stated in the introduction (lines 77-79).

**RC3 response:**

Reviewer comments are marked italic.

In their response they indicated that data from the tekran speciation system has been recently published.

They cite Slemr- that only used GEM data, Wang-who basically lists all the problems with using the Tekran data, Mason who found CEM measured concentrations were 5 time higher than the denuder, and lastly Griggs that does use the speciation system 1130 and 1135 data that should not have been published for the results are very misleading.

We agree with your remarks, although we would like to point out that the discussed comment in RC1 was: "It is important to note that most people do not now use the 1130/1135 unit on the Tekran system and papers with these data are not even being sent out for review."

In response to the aforementioned comment, we have tried to point out that there are in fact a few recent articles submitted for review that use the 1130/1135 unit, regardless of the quality of data or articles. Additionally, we have already mentioned and discussed in the introduction a number of articles that point out the problems of the Tekran atmospheric Hg speciation unit (lines 44-58).